# Identifying the Pressure Points of Acute Cadmium Stress Prior to Acclimation in *Arabidopsis thaliana*

**DOI:** 10.3390/ijms21176232

**Published:** 2020-08-28

**Authors:** Jana Deckers, Sophie Hendrix, Els Prinsen, Jaco Vangronsveld, Ann Cuypers

**Affiliations:** 1Centre for Environmental Sciences, Hasselt University, Agoralaan Building D, 3590 Diepenbeek, Belgium; jana.deckers@uhasselt.be (J.D.); sophie.hendrix@uhasselt.be (S.H.); jaco.vangronsveld@uhasselt.be (J.V.); 2Department of Biology, University of Antwerp, Groenenborgerlaan 171, 2020 Antwerpen, Belgium; els.prinsen@uantwerpen.be

**Keywords:** *Arabidopsis thaliana*, cadmium, acute responses, glutathione, hydrogen peroxide, 1-aminocyclopropane-1-carboxylic acid, ethylene, oxidative challenge

## Abstract

The toxic metal cadmium (Cd) is a major soil pollutant. Knowledge on the acute Cd-induced stress response is required to better understand the triggers and sequence of events that precede plant acclimation. Therefore, we aimed to identify the pressure points of Cd stress using a short-term exposure set-up ranging from 0 h to 24 h. Acute responses related to glutathione (GSH), hydrogen peroxide (H_2_O_2_), 1-aminocyclopropane-1-carboxylic acid (ACC), ethylene and the oxidative challenge were studied at metabolite and/or transcript level in roots and leaves of *Arabidopsis thaliana* either exposed or not to 5 µM Cd. Cadmium rapidly induced root GSH depletion, which might serve as an alert response and modulator of H_2_O_2_ signalling. Concomitantly, a stimulation of root ACC levels was observed. Leaf responses were delayed and did not involve GSH depletion. After 24 h, a defined oxidative challenge became apparent, which was most pronounced in the leaves and concerted with a strong induction of leaf ACC synthesis. We suggest that root GSH depletion is required for a proper alert response rather than being a merely adverse effect. Furthermore, we propose that roots serve as command centre via a.o. root-derived ACC/ethylene to engage the leaves in a proper stress response.

## 1. Introduction

Under the current circumstances, where the growing population is exceeding the global food supply, arable land is becoming sparse [1]. Soil pollution is putting even more restrictions on the availability of qualitative agricultural land. Trace metallic elements, like cadmium (Cd), significantly contribute to this pollution problem, as they are phytotoxic and pose risks to human health via the bioaccumulation in our food chain [2]. The study of short-term plant responses allows for the identification of the pressure points of a certain stressor and the early challenges that plants face prior to acclimation. Understanding the early stress-induced responses will help improve plant acclimation itself, allowing plants, and in particular crops, to reach their full potential even in suboptimal environments. The latter can be achieved by means of biotechnological and agro-ecological approaches which encompass a.o. genetic modifications and application of soil amendments, respectively. 

Cadmium phytotoxicity mainly arises from its bioavailability and chemical similarity to essential elements like zinc, calcium and iron, enabling Cd to hitchhike along transporters for essential elements [3,4]. This results in disturbance of the homeostasis of these elements and in their displacement by Cd in biomolecules, rendering them inactive and, at the same time, freeing up redox-active metals like iron [3,5,6,7]. These redox-active metals directly stimulate reactive oxygen species (ROS) production, while Cd increases ROS production indirectly, via the stimulation of pro-oxidants like NADPH oxidases and the deprivation of the anti-oxidative system [8]. The anti-oxidative metabolite glutathione (GSH) is one of the prominent defence molecules in the responses to Cd stress and is synthesised in two ATP-dependent steps [9,10,11]. Firstly, glutamate is combined with cysteine, which is catalysed by glutamate cysteine ligase (GSH1) to produce γ-glutamylcysteine (γ-EC) [12]. Next, the addition of glycine is catalysed by GSH synthetase (GSH2) to form GSH [13,14]. The nucleophilic nature of the central thiol group enables GSH and its oligomers, termed phytochelatins (PCs), to chelate Cd and sequester it into the vacuole [15]. Concurrently, GSH serves to neutralise ROS, like hydrogen peroxide (H_2_O_2_), directly but mainly through the ascorbate (AsA)-GSH cycle [16,17]. In this cycle, the NADPH-dependent enzyme glutathione reductase (GR) serves to maintain the reduced GSH pool [18]. Moreover, Mhamdi et al. (2010) showed that GR encoded by the *GLUTATHIONE REDUCTASE 1* (*GR1*) isoform is crucial in the metabolism of H_2_O_2_ [19]. In the apoplastic space, however, it is rather unlikely that GSH functions as major anti-oxidant due to its relatively low abundance [20]. Nevertheless, it is clear from the literature that apoplastic GSH and its recycling fulfil other important roles that need further consideration [20,21,22,23,24]. The recycling of extracellular GSH (eGSH) is accomplished by the activity of γ-glutamyl transpeptidase (GGT) encoded by *γ-GLUTAMYL TRANSPEPTIDASE 1* (*GGT1*), that catabolises eGSH into its constituent amino acids [24]. The hydrolysis of eGSH and glutathione S-conjugates enables the recovery of GSH intracellularly [24,25]. Furthermore, it is proposed that GGT plays a role in the redox control of the apoplastic space and serves to mitigate oxidative stress as a result of an unbalanced ROS production [20,24,26]. It is a well-known fact that ROS cannot simply be considered as detrimental compounds, as they often fulfil a signalling role [27]. Hydrogen peroxide, especially, is considered to be a central component of signal transduction due to its stability and ability to cross membranes [28]. Because of this double-edged sword, GSH does not serve merely to detoxify H_2_O_2_, but it is also key in the fine-tuning of H_2_O_2_-dependent signalling responses [29,30].

As demonstrated by Jozefczak et al. (2014), root GSH levels became strongly depleted upon 2 h of Cd (5 µM) exposure in hydroponics, which is attributable to the fact that GSH is allocated to PC synthesis [10]. This impacts the anti-oxidative capacity of GSH in the early responses to Cd stress. Moreover, the depletion of such a ubiquitous and considerable anti-oxidant will most likely trigger specific signalling events that define the acute responses and acclimation to Cd stress. Besides alterations in the GSH pool and H_2_O_2_ signalling, other components like phytohormones are key in the responses to environmental stresses. The important stress hormone ethylene was already demonstrated by Schellingen et al. (2015) to serve as key regulator in the responses to Cd stress [31]. More specifically, ethylene production and signalling are required for the stimulation of leaf GSH metabolism under Cd stress and stimulation of ROS-generating NADPH oxidases in general [31,32,33,34]. Ethylene production is known to increase under Cd stress and mainly relies on the transcriptional upregulation and post-transcriptional stabilisation of 1-aminocyclopropane-1-carboxylic acid (ACC) synthase (ACS) isoforms ACS2 and ACS6 [35]. These isozymes catalyse the conversion of S-adenosylmethionine (SAM), derived from methionine, to ACC, the direct precursor of ethylene [36,37]. It is known that ACS2 and ACS6 are targeted via the transcription factor WRKY33, by mitogen-activated protein kinase 3 (MPK3) and by MPK6, which, in turn, are phosphorylated by the oxidative-signal inducible 1 (OXI1) kinase, that becomes induced by H_2_O_2_ [31,38,39]. These findings demonstrate that different signalling pathways are strongly intertwined in a complex network that defines the outcome of stress responses and acclimation thereafter.

In this study, a time-course analysis of different key regulators was conducted in order to unravel the sequence of events in terms of acute Cd-induced responses. The use of a short-term exposure set-up, ranging from 0 h to 24 h of exposure, allowed for the identification of pressure points prior to acclimation. The identification of these pressure points is highly required in order to understand the hurdles a plant needs to overcome before reaching acclimation and contributes to the bigger picture in order to understand the process of acclimation to Cd stress.

## 2. Results

Focussing on acute Cd-induced responses, both roots and leaves were separately studied during a short exposure time frame ranging from 0 h to 24 h. The use of an optimised hydroponic cultivation system allowed the controlled exposure of *A. thaliana* wild-type (WT) plants to a sublethal and environmentally relevant Cd concentration of 5 µM [40,41].

### 2.1. Cadmium-Induced Growth Responses and Cadmium Accumulation

Fresh weight (Figure 1), dry weight (Appendix A) and Cd concentration (Table 1) were compared between Cd-exposed and unexposed WT plants within the short exposure time frame. Acute exposure did not have a negative impact on the root and leaf fresh weight and dry weight of WT *A. thaliana* plants (Figure 1 and Appendix A). Cadmium uptake and translocation, however, had an early onset, which was reflected by the significant increase in Cd concentrations observed in both organs 2 h after the start of exposure (Table 1).

Generally, Cd concentrations were higher in roots than in leaves. Both root and leaf Cd concentrations increased significantly in a time-dependent manner and especially after 24 strong increases were observed in both organs (Table 1). Similar to the Cd concentrations in both organs, the translocation of Cd from root to shoot increased time-dependently, and after 24 h a translocation factor of almost 50% was reached (Table 1). This means that 50% of the Cd taken up by the roots was translocated to the aerial part of the plant. In our study, Cd translocation had an early onset and, even though Cd translocation is important in the detoxification of Cd, PC synthesis is the major determinant of Cd sensitivity in general [42].

### 2.2. Glutathione as Chelator and Anti-Oxidant

The chelation of Cd by PCs followed by vacuolar sequestration is a well-characterised mechanism of Cd detoxification [15]. In this study, the allocation of GSH to PC synthesis was reflected by a significant depletion of GSH in the roots after 2 h of exposure (Figure 2), which was also demonstrated by Jozefczak et al. (2014) [10]. After 4 h of exposure, GSH depletion became even more pronounced and the root GSH levels dropped below 40% of GSH levels in control plants. After 6 h of exposure, a turning point was observed and root GSH levels revealed an increasing trend. While GSH concentrations were still significantly decreased after 6 h of exposure, they were fully recovered to control levels after 24 h (Figure 2).

This recovery of root GSH levels starting from 6 h of exposure coincided with a significantly increased expression of GSH metabolism genes (*GSH1, GSH2* and *GR1*) (Table 2). In leaves of Cd-exposed plants, the depletion in GSH levels was not observed (Figure 2). In fact, GSH levels remained unchanged, and the expression of metabolism genes was higher after 24 h of exposure (Table 2). Even though total root GSH concentrations fluctuated to a great extent throughout the short-term Cd exposure, the percentages of reduced GSH remained tightly controlled. More specifically, over 90% of the GSH pool occurred in its reduced form, and this was maintained at each time point in both organs (Appendix A). Furthermore, only limited fluctuations in GR activity were observed (Table 3). A limited decrease in GR activity occurred after 2 h of Cd exposure in the roots, and a significant increase in activity became apparent after 24 h of exposure. In the leaves, GR activity was significantly elevated after 6 h and 24 h of Cd exposure (Table 3).

As reported in literature, no GR activity was detected in the apoplastic space, and GSH recycling was carried out by *GGT1*-encoded GGT [25]. Because the concentration of GSH in the apoplast is relatively low, AsA is more likely to fulfil the role as major anti-oxidant [20]. However, eGSH and its recycling by GGT have been implemented in the mitigation of oxidative stress and were demonstrated to be important in the modulation of stress signalling [21]. In our study, a strong and early transcriptional induction was observed for GGT1 at the root level (Table 2).

### 2.3. ROS Signalling and Oxidative Challenge under Acute Cd Stress

In roots, the transcriptional profile of the ROS-producing NADPH oxidases Respiratory Burst Oxidase Homologues D and F (*RBOHD* and *RBOHF*, Table 4) showed an early and significant induction after 2 h, which remained elevated throughout the entire 24 h Cd exposure period. A similar expression pattern was observed for *RBOHC* but its induction had already disappeared after 24 h (Table 4). Oxidative stress, as indicated by the induction of oxidative stress markers [43], had a delayed onset at the root level and occurred to a smaller extent as compared to the leaves. More specifically, oxidative stress markers and related genes (i.e., *ZINC FINGER OF ARABIDOPSIS THALIANA 12* (*ZAT12*) and *REDOX-RESPONSIVE TRANSCRIPTION FACTOR 1* (*RRTF1*)) were significantly induced or, at least, showed an increasing trend but only upon 24 h of Cd exposure (Table 4).

Although *RBOHC* is known to be very low abundant in leaves under control conditions, a strong increase in transcripts was already observed in the leaves after 4 h of exposure, and a fifty-fold increase was even seen after 24 h of exposure (Table 4) [44]. The gene expression levels of both *RBOHD* and *RBOHF* in the leaves were only significantly elevated after 24 h of Cd exposure. Both peaks in the induction of the transcript levels of these ROS-generating enzymes were accompanied by a certain extent of oxidative challenge, as reflected by the higher expression of the oxidative stress hallmark genes [43]. More specifically, after 4 h of exposure, transcript levels of *AT1G05340* and *AT1G19020* were increased and coincided with the induction of *RBOHC* in the leaves (Table 4). After 24 h, oxidative challenge occurred to a larger extent, as reflected by a simultaneous and strong induction of all stress markers and oxidative challenge-related genes (Table 4). 

While Cd-induced oxidative challenge originates from an indirect rise of general ROS levels, our attention was drawn towards H_2_O_2_, which is often put forward as a prominent signalling molecule [28,45]. In both roots and leaves, H_2_O_2_ levels were significantly enhanced at 24 h of Cd exposure (Figure 3a), which occurred in parallel with an increase in transcript levels of NADPH oxidases and oxidative stress-related genes (Table 4). As H_2_O_2_ levels are often modulated by GSH, and because free GSH levels fluctuated strongly in our exposure set-up, we considered relative H_2_O_2_ changes in relation to the Cd-induced changes in GSH levels (Figure 3b). This allowed us to obtain an integrative view of the redox status in both organs. The highest ratios were observed in the roots between 2 h and 6 h of exposure, with a peak observed after 4 h. Even though still significantly elevated, the ratio seemed to stabilise after 24 h in the roots. In the leaves, no significant changes were observed (Figure 3b).

### 2.4. Ethylene-Related Signalling and Stress Responses

The stress hormone ethylene is known to be a key regulator of plant responses to metal stress [46]. Its vital role in the responses to Cd stress was already brought forward by Schellingen et al. (2014 and 2015) [31,35]. These studies demonstrated that Cd exposure stimulates the synthesis of ethylene that, in turn, mediates the Cd-induced responses via, for example, the stimulation of GSH biosynthesis and metabolism [31,35]. In this study, the transcriptional profile of ethylene biosynthesis and responsive genes was considered together with the production of ACC, its direct precursor (Table 5 and Figure 4). The latter, which is the intermediate between SAM and ethylene, is often put forward as a short- and long-distance signalling molecule both dependent and independent of ethylene [47]. In this study, both ACS isoforms were increased at the transcript level under Cd stress in the roots. However, *ACS6* was already significantly induced starting from 4 h of exposure, followed by *ACO2* and *ACO4* after 6 h, and after 24 h *ACS2* was also significantly higher. All transcripts remained significantly elevated up to 24 h of exposure (Table 5). Furthermore, the significantly increased expression of *ERF1*, starting from 6 h, hints towards active ethylene signalling at the root level. This ethylene response established at the transcript level was preceded by the significant induction of *MPK3* and *MPK6* after 2 h and *OXI1* and *WRKY33* after 4 h (Table 5). The transcription factor WRKY33 acts downstream of the MPK3/6 pathway.

Similarly, in the leaves of Cd-exposed plants, *ERF1* was also significantly increased after 6 h and 24 h and coincided with an increase in *OXI1* transcripts, which was preceded by a significant upregulation of *MPK6* (Table 5).

Considering the ethylene biosynthesis-related genes, overall strong and significant increases were observed after 24 h of exposure for all genes considered (Table 5). Concerning free ACC concentrations, responses strongly differed between roots and leaves. In roots, free ACC concentrations were already significantly elevated compared to the control levels starting from 2 h of Cd exposure and further increased towards 24 h (Figure 4). A delayed response was observed for the leaves as non-conjugated ACC levels were only significantly increased after 24 h of exposure, albeit to a very large extent (Figure 4).

## 3. Discussion

Plants possess a great plasticity and adaptive potential, enabling them to cope with a broad range of environmental stresses and acclimate to changing environments. The process of acclimation encompasses homeostatic adjustments and results in newly established equilibria [48,49]. However, acclimation is typically preceded by a stress response, which generally occurs within a time frame of seconds to days, and most often leads to a temporary suboptimal performance [48]. The study of acute responses to Cd exposure, within the period of 0 h to 24 h, allows us to identify the pressure points of Cd stress before new equilibria are reached. Knowledge concerning such pressure points and acute responses is required to improve our understanding of the triggers and sequence of events that precede acclimation. The obtained knowledge can contribute to the improvement of plant acclimation, enabling plants to reach their full capacity even under stressful conditions. This trait is highly required in the current conditions, since non-polluted arable land is becoming sparse. 

While the dual role of GSH under Cd stress, as a chelator and anti-oxidant, was already touched upon by Jozefczak et al. (2014), our data further underline the dilemmas plants encounter in their acute responses to environmentally realistic, sublethal (i.e., 5 µM) Cd stress [10,41]. More specifically, the present study further uncovers the Cd-induced trade-offs between (1) GSH as a chelator and anti-oxidative metabolite and (2) ROS signalling and oxidative stress, i.e., an oxidative challenge. From our data, it became clear that these trade-offs were mainly manifested at the root level, especially with regard to GSH, that became strongly depleted within 2 h of Cd exposure (Figure 2). Jozefczak et al. (2014) indicated that this depletion occurred due to the allocation of GSH to its Cd-chelating oligomers, namely phytochelatins [10]. The fact that, in the present study, GSH levels were even more strongly depleted after 4 h adds to our knowledge that the effect on free GSH levels becomes more pronounced and persists at least until 4 h after exposure (Figure 2). Overall, the Cd-induced changes in root GSH concentrations fit the typical stress response curve described by Lambers et al. (1998) [48]. First of all, the initial alarming phase becomes visible by the rapid depletion of GSH followed by the restitution phase, which is established between 6 h and 24 h, as a full recovery to control levels occurred within this time frame (Figure 2). As shown previously, overcompensation by increased GSH levels does not occur at the root level under 5 µM of Cd, at least not after 24 h, 48 h and 72 h [10]. At the leaf level, even though Cd was translocated early on and the leaf Cd concentrations were significantly elevated from 2 h onwards (Table 1), no changes in GSH concentrations were observed within the considered 24 h time frame (Figure 2). However, it is known that leaf GSH levels significantly increase after 48 h and 72 h of exposure to 5 µM Cd [31]. As reviewed by Tausz et al. (2004) and Zagorchev et al. (2013), an increase in GSH concentrations is often observed as an acclimatory response to a range of stresses and hints towards a better stress resistance and a new steady-state [49,50]. Note, however, that increased GSH levels, exogenously applied or transgenically enhanced, do not necessarily imply an improved tolerance, especially in the case of Cd stress [51,52]. The fact that the manipulation of GSH levels can lead to increased Cd sensitivity underlines the fine-tuning that is required for proper acclimation and emphasizes the importance of the alarming phase, provoked by the rapid and strong GSH depletion at the root level (Figure 2). Moreover, our data indicate that the depletion and recovery of GSH levels have a relatively fast nature, which further confirms the fact that the pressure points of stress factors are often overlooked when considering longer exposure time frames and underlines the importance of monitoring stresses at different time points. In addition, the fact that responses strongly differ between roots and leaves points out that at least both organs need to be considered when studying plant stress responses and acclimation, especially when the stress is (partly) propagated via the root system. 

Even though several studies have indicated that a rapid and transient depletion of GSH occurs after exposure to excess metal concentrations, little is known about the impact of this event on the plant’s responses [10,53,54]. Indeed, depletion of this prominent anti-oxidant could lead to an oxidative challenge at the root level as its anti-oxidative capacity is largely impaired. However, the extent to which this event is detrimental or, on the contrary, contributes to stress signalling—and, ultimately, plant acclimation—remains unclear. Alterations of the GSH pool and generation of the prominent ROS signalling molecule H_2_O_2_ are both central components of stress-induced signal transduction and often act in concert [29,45]. The ratio between oxidising H_2_O_2_ and the important anti-oxidant GSH allows us to obtain an integrative view of the Cd-induced redox changes and shows that H_2_O_2_ levels in the roots are most strongly elevated in relation to GSH early on (Figure 3b). Previous studies have indicated that the changes in GSH status are rather a modulator of the stress-induced increases in H_2_O_2_ than merely a passive result [29,30]. In our study, the Cd-induced H_2_O_2_ increases are modulated by the depletion of the GSH pool and are not influenced by changes in its redox state, because the redox state of the GSH pool was not affected by Cd exposure and the percentage of reduced GSH remained tightly controlled above 90% (Appendix A). Consistent with these findings, it was shown by Schnaubelt et al. (2015) that buthionine sulfoximine (BSO)-induced depletion of the root GSH level did not necessarily impact its redox state [30]. Moreover, lowered GSH levels counteract its oxidation [29]. As GR is key in the recycling of the oxidised GSSG back to its reduced form, an increased GR activity may explain this tight control. However, in our study, GR activity in the roots was only significantly increased after 24 h of exposure (Table 3). Therefore, it can be concluded that its activity is not contributing to maintain a reduced GSH pool at the early time points when GSH becomes depleted (Figure 2). As shown by several studies, *GSH1* and *GSH2* are typically induced upon Cd exposure [9,10,55,56,57]. In our study, their transcriptional induction coincided with the restoration of GSH levels observed after 6 h and 24 h at the root level (Figure 2). The catalyser of the first and rate-limiting step of GSH biosynthesis, *GSH1,* is regulated at several levels. For example, GSH itself is known to have a negative impact on its own production by inhibition of GSH1 activity [57]. Hence, the GSH-depleted conditions (Figure 2) in our study favour an increased activity of GSH1 in the roots. Moreover, at the transcript level, the transcription factor ZAT6 is known to stimulate *GSH1* transcription and its own expression is enhanced upon Cd exposure, which was also observed in our study [56]. In general, this study is in agreement with the conclusion drawn by Han et al. (2013) that the plant cell redox status is configured in such a way that depletion inhibits GSH oxidation and strong changes in GSH concentration are sufficient to alter the cell’s redox potential and drive GSH accumulation [29]. 

It is clear that intracellular GSH is a major modulator of stress responses and the acclimation process thereafter. Noteworthy, however, is the eGSH residing in the apoplastic space, which is largely regulated by the γ-glutamyl cycle [24,25]. The fine-tuning of the apoplastic GSH content by this cycle serves in redox, balancing the apoplastic space and recovery of GSH—or, more precisely, its constituent amino acids—into the cell. The driving force of eGSH degradation is the apoplastic GGT enzyme encoded by *GGT1*, which catalyses the transfer of the γ-glutamyl group of GSH to a range of acceptors like water or another amino acid [24]. Considering there is no mechanism to reduce extracellular GSSG, this enzyme prevents the accumulation of GSSG in the apoplastic space, mitigating oxidative stress [24]. Even though *GGT1* is most strongly expressed in the leaves of *A. thaliana*, our data show that the Cd-induced *GGT1* upregulation is more pronounced and occurs faster in the root system (Table 2) [25]. Accordingly, enzyme-histochemical analyses showed that GGT activity was very intense in root tips of *Hordeum vulgare* and *Zea mays* [26,58]. Furthermore, Uzilday et al. (2018) observed a strong induction of *GGT1* under endoplasmic reticulum stress, a stress that is also known to be evoked by short-term Cd exposure [59,60]. It has been suggested that the γ-glutamyl cycle serves to link the environment to the plant cell and may provide a way to transfer redox information between the apoplast and the symplast [20]. Other key components known to be involved in the apoplastic redox regulation that bridge the extracellular and intracellular space are NADPH oxidases. In our study, a similar transcriptional profile was observed for *RBOHC, D* and *F*, which coincided with *GGT1* expression (Table 2 and Table 3). The Cd-induced transcription of these prominent NADPH oxidase isoforms (Table 3) hints at an increased production of superoxide and subsequently H_2_O_2_ in the roots. An augmented H_2_O_2_ production, as observed in our study (Figure 3), could lead to oxidation of the apoplastic GSH pool and activation of the γ-glutamyl cycle. In summary, these data point towards a redox-related signalling event that is in full practice upon 2 h of Cd exposure and persists at least up to 24 h of exposure. Furthermore, as shown by Tolin et al. (2013), apoplastic GGT is an important modulator of the redox response, since the knockout of *GGT1* leads to a constitutive “alert response” even in absence of environmental stimuli [20,21]. Therefore, it can be suggested that GGT encoded by *GGT1* also functions as an important modulator in the redox sensing and signalling under Cd stress. 

In our study, only a small subset of oxidative stress markers was transcriptionally induced in the root, and only after 24 h of exposure (Table 4) [43]. This is in line with our suggestion that the early alterations observed in the H_2_O_2_/GSH at the root level are required for a proper signalling response under Cd stress, rather than having merely a detrimental oxidative stress effect. In addition, the fact that the transcriptional profile indicating oxidative stress markers in the roots was only induced to a limited extent and delayed, is in line with the findings of Schnaubelt et al. (2015). They suggested that GSH depletion evokes a very specific response as the transcriptome of the *root meristemless 1-1* (*rml1-1*) mutant, harbouring only 2.7% of WT GSH levels. This response is different from that of the *catalase 2* (*cat2-1*) mutant and lacks an induction of the oxidative stress markers, which might be explained by the absence of a change in the GSH/GSSG ratio [30]. Another important modulator of ROS signalling is the transcription factor *RRTF1* that was transcriptionally induced in both roots and leaves but only upon 24 h (Table 4). Matsuo et al. (2015) demonstrated that the expression of *RRTF1* is stimulated by ROS and that RRTF1 itself is responsible for the amplification of ROS generated by a stressor that perturbs basal ROS levels [61]. Additionally, one of its target genes, *ZAT12,* also became significantly upregulated upon 24 h of Cd exposure (Table 4), which in its turn stimulates the transcription of, for example, *RBOHD*, that was increased after 24 h as well (Table 3) [61]. Both are implemented in the regulation of ROS signalling under unfavourable conditions, and in this case their upregulation indicates ROS amplification to possibly intensify responses after 24 h when root GSH levels are stabilised. Indeed, at this time point H_2_O_2_ levels were significantly increased in both root and leaves independently of GSH (Figure 3a). In this way, the Cd-induced signalling responses might be redirected away from GSH-dependent redox sensing, as root GSH levels are restored to control levels at this later time point (Figure 2) and can no longer serve as a redox signal.

Our data indicate that the GSH-related leaf responses are delayed and less pronounced in comparison to the roots (Figure 2 and Table 2). This is plausible, since the roots are in direct contact with the Cd-containing nutrient solution. However, our data show that Cd is translocated early on to the aerial parts, leading to significantly higher Cd concentrations in the leaves compared to the control already after 2 h of exposure (Table 1). Nevertheless, no GSH depletion was observed in the leaves (Figure 2). As mentioned before, Cd-induced GSH depletion is caused by the allocation of GSH to PC synthesis, a process that is also known to occur in leaves after 24 h of exposure to 5 µM Cd [10]. Hence, since leaf GSH levels are not negatively affected (Figure 2), leaf signalling responses possibly directly shift to the GSH-independent signalling response, as described above. Our data indicate that Cd-induced leaf GSH stimulation serves to buffer the impact of stresses at the leaf level and that GSH fulfils a protective role rather than a signalling role. This implies, however, that other components are responsible for stress signalling in the leaves in order to reach acclimation.

It is clear from our study that the early depletion in root GSH levels does not stand alone. More specifically, the production of the ethylene precursor and important signalling molecule ACC was already significantly higher from 2 h of exposure onwards at the root level (Figure 4). As demonstrated by Schellingen et al. (2015), ROS signalling is integrated into the signalling cascade that precedes Cd-induced ethylene biosynthesis by increasing *OXI1* expression, which, in its turn, activates MPK3 and MPK6 [31,38]. These kinases target ACS2 and ACS6, leading to an increase in their half-life and a stimulation of their gene expression [62,63,64]. In our study, both *MPK3* and *MPK6* seem to function early in the root responses to Cd stress, and their transcriptional induction (Table 5) collides with the higher root ACC concentrations (Figure 4). Moreover, it is known that, in *A. thaliana*, *OXI1* gene expression and kinase activity are induced upon exposure to a broad range of H_2_O_2_-generating stimuli [38]. One such stimulus could, for example, originate from RBOHC, and indeed the knockout of *RBOHC* leads to a decreased induction of *OXI1* in the roots of *A. thaliana* [44]. Correspondingly, in our study, the induction of *OXI1* (Table 5) peaks in concert with the highest H_2_O_2_/GSH ratio observed in the roots (Figure 3b) and possibly leads to the activation of the aforementioned signalling cascade, with the transcriptional induction of *ACS2* and *ACS6* as an end result (Table 4). The fact that the ACC concentration (Figure 4) in the roots was already significantly higher after 2 h of exposure, and therefore preceded the transcriptional induction of *ACS2* and *ACS6* (Table 4), suggests a rapid activation at the protein level, which later on is extended to the transcript level.

Additionally, ethylene and GSH are strongly intertwined in the responses to Cd stress. It was shown by Schellingen et al. (2015) that leaf GSH stimulation under Cd stress depends on ethylene signalling [31]. More precisely, the considered ethylene insensitive *ein2-1* mutants proved unable to increase their leaf GSH levels upon Cd exposure. This also became apparent at the transcript level, as the induction of the GSH metabolism genes was abolished in these mutants [31]. It is clear from our point of view that, at least at the transcript level, ethylene signalling (Table 4) precedes the induction of *GSH1, GSH2* and *GR1* in the leaves (Table 2). It should be noted, however, that leaf ethylene signalling, assessed by the induction of ethylene-responsive genes such as *ERF1* (Table 5), precedes the higher leaf ACC concentrations observed after 24 h of Cd exposure (Figure 4). Therefore, we suggest that the transcriptional induction of *ERF1* is not a result of a de novo ethylene synthesis originating from the leaf, but of ethylene produced at the root level that evokes responses in the leaves. However, cross-talk with other phytohormones, like jasmonate, should not be neglected. Nevertheless, studies found that stress-induced *ERF1* expression was strongly diminished in the leaves of the ethylene biosynthesis double mutant *acs2-6*, which further corroborates that ethylene is largely responsible for the induction of *ERF1* in the leaves [35,63]. Indeed, at the level of the root, ethylene synthesis, or at least ACC concentration (Figure 4), was already significantly increased upon 2 h of exposure. Therefore, the root system might serve as a command centre and delivers stress-related signals, like ethylene, to the leaves. The latter will engage in an optimal response that becomes apparent by the stimulation in leaf GSH biosynthesis rather than GSH depletion, which occurs in the roots after similar Cd concentrations are encountered (Figure 2).

## 4. Materials and Methods

### 4.1. Plant Culture, Cadmium Treatment and Sampling

Seeds of wild-type *Arabidopsis thaliana* plants (Columbia background) were surface-sterilised and incubated in the dark for 3 nights at 4 °C. Seedlings were grown in a hydroponic culture system using purified sand as a substrate and a modified Hoagland nutrient solution [40,41]. Growth conditions were set at 65% relative humidity under a photoperiod of 12 h with day/night temperatures of 22 °C and 18 °C, respectively. The photosynthetically active radiation of sunlight was simulated by providing a combination of blue, red and far-red light (Philips Green-Power LED modules) with a photosynthetic photon flux density of 170 μmol m^−2^ s^−1^ at the rosette level. After 3 weeks of growth, wild-type plants were exposed to 0 (control) or 5 μM Cd via addition of CdSO_4_ to the nutrient solution. Plants were harvested at different points (0 h, 2 h, 4 h, 6 h and 24 h) after the start of exposure. Root and rosette fresh weight were determined and samples were snap frozen in liquid nitrogen and stored at −70 °C unless stated otherwise.

### 4.2. Quantification of Root and Rosette Cd Concentrations

Harvested leaves were rinsed with distilled water to remove any residual metals. Roots were submerged for 15 min in 10 mM Pb(NO_3_)_2_ at 4 °C, to exchange surface-bound metals, and rinsed in distilled water. Prior to analysis, samples were oven-dried at 80 °C and digested in HNO3 (70–71%) and HCl (37%). Cadmium concentrations were determined via inductive coupled plasma-optical emission spectrometry (ICP-OES; Agilent Technologies 700 Series, Santa Clara, CA, USA). For reference purposes, blank (HNO_3_) and standard (trace elements in spinach, 1570a, Standard Reference Material) samples were included.

### 4.3. Gene Expression Analysis

In order to study Cd-induced responses at the transcriptional level, gene expression analysis was carried out. Root and leaf samples were disrupted under frozen conditions by shredding using two stainless steel beads and the Retsch Mixer Mill MM400 (Retsch, Haan, Germany). Isolation of RNA from the disrupted sample tissue was conducted using the RNAqueous^®^ Kit (Thermo Fisher Scientific, Waltham, MA, USA) according to the manufacturer’s instructions. By use of the Nanodrop^®^ ND-1000 Spectrophotometer (Thermo Fisher Scientific), RNA purity and concentration were spectrophotometrically determined. Finally, RNA integrity was analysed using gel electrophoresis, and samples were stored at −70 °C. After RNA isolation, cDNA was synthesised from equal RNA inputs (1 μg) via reverse transcription using the Primescript^TM^ RT reagent Kit (Takara Bio Inc., Kusatsu, Japan). To remove residual genomic DNA, the RNA samples (1 μg) were pre-treated with DNase using the TURBO DNA-*free*^TM^ Kit (Ambion). Prior to storage at −20 °C, the cDNA was ten-fold diluted using a 1/10 Tris-EDTA buffer (1 mM Tris–HCl, 0.1 mM Na_2_-EDTA, pH 8.0). 

Using quantitative real-time PCR (qPCR), the expression levels of the genes of interest (GOIs; Appendix A) were determined. Reactions were carried out in a 96-well plate with the 7500 Fast Real-Time PCR System (Applied Biosystems, Thermo Fisher Scientific, Foster City, CA, USA) using the Fast SYBR^®^ Green Master Mix (Applied Biosystems, Thermo Fisher Scientific) according to the manufacturer’s protocol. Amplification was performed under universal cycling conditions (20 s 95 °C, 40 cycles of 3 s at 95 °C and 30 s at 60 °C), followed by the generation of a dissociation curve to verify amplicon specificity. The reactions contained 2 µL of the diluted cDNA template (or RNase-free H_2_O for the “no template controls”), 5 µL 2× Fast SYBR^®^ Green Master Mix (Applied Biosystems, Thermo Fisher Scientific) and forward and reverse primers (300 nM each, unless mentioned otherwise in Appendix A), in a total volume of 10 µL. Based on the obtained Cq values, the relative gene expression level of the GOIs was determined using the 2^−ΔCq^ method. Technical variation was corrected for by normalisation with the geometric average of at least three reference genes (Appendix A) that were selected from 10 candidate reference genes based on the GrayNorm Algorithm [65]. Gene-specific forward and reverse primers were designed and optimised via the Primer3 software. Primer specificity was verified in silico using Blast (http://www.arabidopsis.org/Blast/index.jsp). In order to guarantee an optimal reaction efficiency, primer efficiency (E) was evaluated on a standard curve using a two-fold dilution series of a pooled sample and verified to be between 90% and 110%. All gene annotations, primer sequences and primer efficiencies are shown in Appendix A. The qPCR parameters according to the Minimum Information for publication of qPCR Experiments (MIQE) guidelines are shown in Appendix A [66].

### 4.4. Glutathione Concentration

Both oxidised and reduced GSH were spectrophotometrically determined in root and leaf samples according to the plate reader method described by Queval and Noctor (2007) and modified by Jozefczak et al. (2014) [10,67]. Additionally, minor modifications were carried out concerning the extraction method. Frozen samples (75 mg) were ground using two stainless steel beads and the Retsch Mixer Mill MM400 (Retsch). Samples were further homogenised by adding 200 mM HCl (6.66 mL mg^−1^ leaf fresh weight; 9.4 mL mg^−1^ root fresh weight) and vortexing. After centrifugation (10 min, 16,000× *g*, 4 °C), the pH of the samples was adjusted to 4.5 using 200 mM NaOH, and the samples were kept at 4 °C throughout the procedure. The assay relied on the GSH-dependent reduction of 5,5-dithiobis (2-nitro-benzoic acid) (DTNB, 600 µM), which was monitored at 412 nm in a FLUOstar Omega microplate reader (BMG Labtech, Ortenberg, Germany). Formed GSSG or GSSG present in the sample was reduced by glutathione reductase (1 U mL^−1^) in the presence of NADPH (500 mM). The absorbance rate over time was proportional to the GSH concentration in the samples, which was determined using a GSH standard curve. In order to measure GSSG, GSH was first complexed by incubating the samples with 1% 2-vinylpyridine (2-VP) (30 min, room temperature). Complexed GSH and 2-VP were removed by centrifuging the samples twice (10 min, 16,000× *g*, 4 °C) prior to the measurement.

### 4.5. Hydrogen Peroxide Measurements

Relative hydrogen peroxide (H_2_O_2_) concentrations were determined in roots and leaves using the Amplex™ Red Hydrogen Peroxide/Peroxidase Assay Kit (Invitrogen, Thermo Fisher Scientific, Carlsbad, CA, USA). Frozen samples were disrupted using two stainless steel beads in the Retsch Mixer Mill MM 400 (Retsch). The extraction was carried out in 500 μL 1× Reaction Buffer, and the samples were shaken continuously at room temperature for 30 min and centrifuged at 12,000× *g* for 5 min. Reactions were performed in a 96-well plate with each well containing 95 μL of a working solution consisting of 100 μM Amplex™ Red and 0.2 U/mL horseradish peroxidase, to which 5 μL supernatant was added. After 30 min incubation in the dark at 30 °C, the samples were excited at 560 nm, and resorufin fluorescence was measured at 590 nm in a FLUOstar Omega microplate reader (BMG Labtech, Ortenberg, Germany).

### 4.6. Determination of Free ACC Content

Frozen root and leaf (200 mg) samples were disrupted using the Retsch Mixer Mill MM 400 (Retsch) and two stainless steel beads. To allow for quantification, [^2^H_4_] ACC (200 pmol, Sigma, St. Louis, MO, USA) was added as an internal standard. After centrifugation (20,817× *g*, 15 min, 4 °C, Eppendorf 5810R, Hamburg, Germany), ACC was extracted using the solid-phase extraction procedure described by Smets et al. (2003) [68]. Samples were derivatised with pentafluorobenzyl (PFB) bromide (Sigma, Saint Louis, MO, USA) and analysed as PFB-bis-ACC by Negative Ion Chemical Ionisation Gas chromatography–mass spectrometry (NICI GC-MS; Quattro micro MS/MS, Waters; electron energy 70 eV, emission 200 μA, extraction 10 V, source 206 μA, GC interface T: 120 °C, CI gas flow 69 mL/min). The GC (WCOT) column was purchased from Varian (CP-Sil 5 C8 Low bleed/MS column, 30 m, 250 μm, film thickness 0.25 μm) using helium as the mobile phase (T gradient 50 °C to 250 °C at 25 °C/min). Corresponding to their pentafluorobenzyl (PFBbis-ACC) derivatives, the following diagnostic transitions were used for Multiple Reaction Monitoring (MRM): 280 > 112 and 280 > 167 for ACC and 284 > 116 and 284 > 167 for D_4_-ACC. Based on the transitions, 280 > 112 and 284 > 116 concentrations were calculated.

### 4.7. Glutathione Reductase Activity Measurements

Root and leaf samples (150 mg) were homogenised under frozen conditions in a 1 mL ice-cold 0.1 M Tris–HCl buffer (pH 7.8) containing 1 mM EDTA, 1 mM DTT and 4% insoluble polyvinylpyrrolidone, using sand and a mortar and pestle. To remove the sand, the homogenate was squeezed through a nylon mesh and centrifuged for 10 min at 20,000× *g* and 4 °C. The glutathione reductase activity, based on the reduction of GSSG using NADPH, was measured spectrophotometrically in the supernatant at 25 °C [69].

### 4.8. Statistical Analyses

Prior to analysis, outliers were determined using the Extreme Studentised Deviate method (GraphPad Software, La Jolla, CA, USA) at significance level 0.05. Statistical analyses were performed in R version 3.3.1 (R Foundation for Statistical Computing, 2016, Vienna, Austria). Data were checked for both normal distribution (Shapiro–Wilk test) and homoscedasticity (Bartlett’s test). In case these assumptions were not met, data were transformed (inverse, square root, logarithm, exponent) to achieve both normality and homoscedasticity of the data. The data were statistically analysed using a student’s *t*-test or one-way ANOVA at significance level 0.05. To correct for multiple comparisons, a post-hoc Tukey–Kramer test was used. If data were not normally distributed and/or homoscedastic, the non-parametric Kruskall–Wallis test was used in combination with the Wilcoxon rank sum test.

## 5. Conclusions

Our study on acute Cd-induced responses has put a subset of pressure points into perspective and elucidated, at least partly, the timing of these short-term responses (Figure 5). It is clear that root GSH depletion under Cd stress is the harbinger of a sequence of events. Moreover, we suggest that, corresponding to other studies, the root GSH depletion is a requirement for a proper Cd-induced alert response and, consequently, optimal acclimation. In this study we further substantiated previous findings that consider GSH as an important modulator of stress-induced H_2_O_2_ increases [29]. As both are central components of stress signalling, often acting in concert, the considered H_2_O_2_/GSH ratio allowed us to obtain an integrative view on the plant’s redox changes. Moreover, we show that, in case of Cd stress, H_2_O_2_ fluctuations are fine-tuned by the overall GSH pool rather than its redox state. The fact that, at the root level, only a subset of oxidative stress markers is upregulated, and this only after 24 h of exposure, further substantiates our suggestion that the early alterations observed in H_2_O_2_/GSH at the root level are required for a proper signalling response under Cd stress, rather than being merely a detrimental effect. The extent to which the redox balance in the apoplastic space is involved in the early responses to Cd exposure and, more specifically, serves in the sensing and propagation of redox signalling, needs further consideration. Finally, we put forward a model wherein the GSH-based alarming phase will help enhance root ethylene synthesis, derived from the rapidly increased ACC concentrations (Figure 5) [31]. This root-derived ethylene and/or ACC might function as a root-to-shoot signal that serves to optimise responses in the leaves, which eventually result in increased leaf GSH levels. This study revealed several aspects that are key in the early responses to Cd stress and serves as a stepping stone to future studies on key regulators of the Cd-induced alert response in view of the subsequent acclimation. The chicken or egg paradox concerning ethylene and GSH has yet to be further elucidated, and our data suggest that an even shorter exposure time frame allows for a clearer view on their interaction.

## Figures and Tables

**Figure 1 ijms-21-06232-f001:**
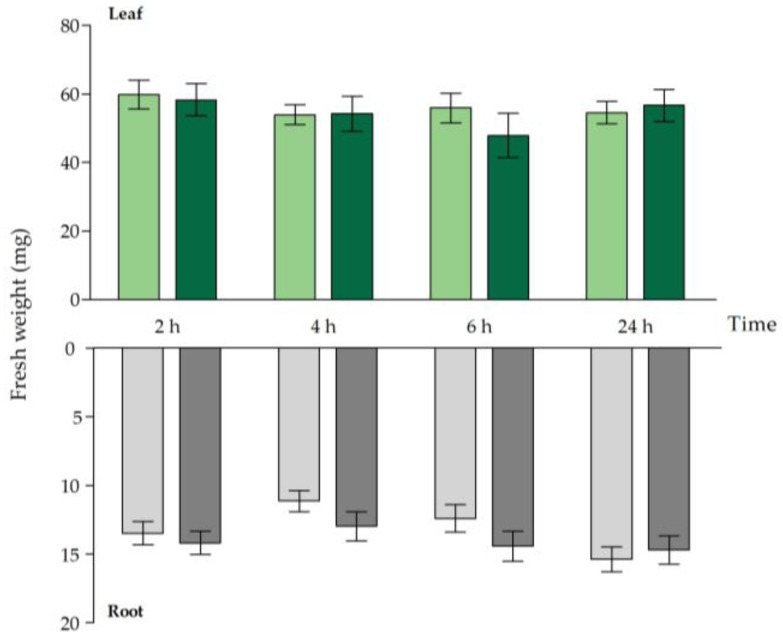
Rosette (green bars) and root (grey bars) fresh weight (mg) of *Arabidopsis thaliana* plants grown under control conditions (0 μM CdSO4, light bars) or exposed to 5 μM CdSO4 (dark bars) during 2 h, 4 h, 6 h and 24 h after 3 weeks of growth. For each time point, data represent the mean ± S.E. of eight biological independent replicates. No significant differences (*t*-test: *p* < 0.05) were observed between control and exposed plants, within each time point.

**Figure 2 ijms-21-06232-f002:**
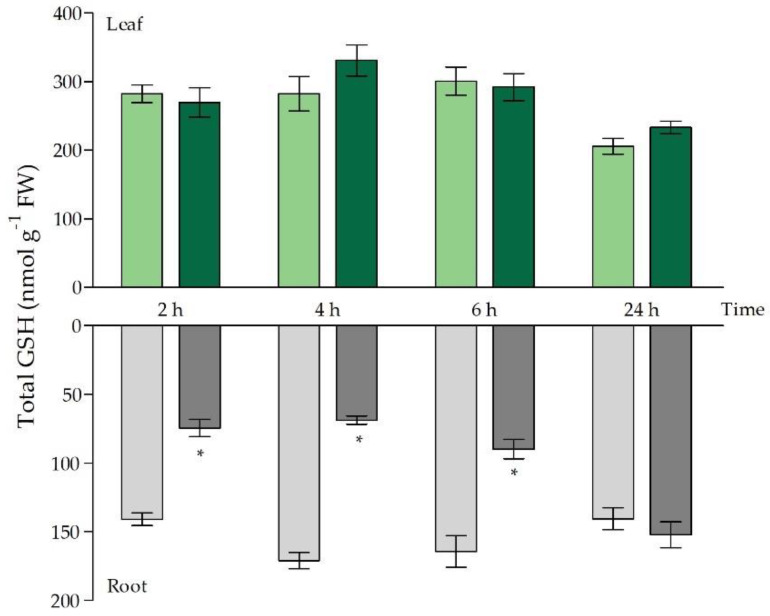
Total glutathione (GSH) concentrations (nmol g^−1^ fresh weight) in leaves (green bars) and roots (grey bars) of *Arabidopsis thaliana* plants grown under control conditions (0 μM CdSO4, light bars) or exposed to 5 μM CdSO4 (dark bars) during 2 h, 4 h, 6 h and 24 h after 3 weeks of growth. For each time point, data represent the mean ± S.E. of four biological independent replicates. Significant differences (*t*-test: *p* < 0.05) between control and exposed plants, within each time point, are indicated with an asterisk (*).

**Figure 3 ijms-21-06232-f003:**
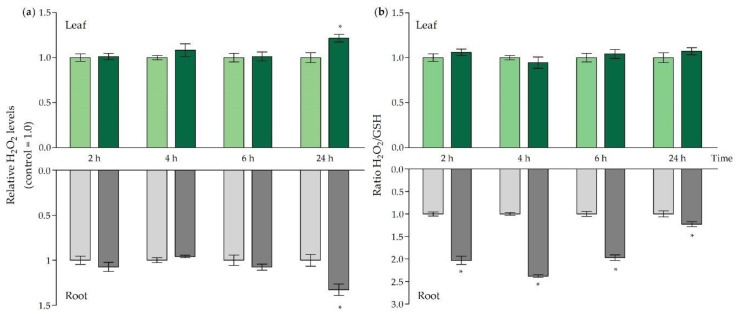
(**a**) Relative hydrogen peroxide (H_2_O_2_) levels and (**b**) H_2_O_2_/GSH ratios in leaves (green bars) and roots (grey bars) of *Arabidopsis thaliana* plants grown under control conditions (0 μM CdSO_4_, light bars) or exposed to 5 μM CdSO_4_ (dark bars) during 2 h, 4 h, 6 h and 24 h after 3 weeks of growth. For each time point, data represent the mean ± S.E. of four biological independent replicates. Significant differences (*t*-test: *p* < 0.05) between control and exposed plants, within each time point, are indicated with an asterisk (*).

**Figure 4 ijms-21-06232-f004:**
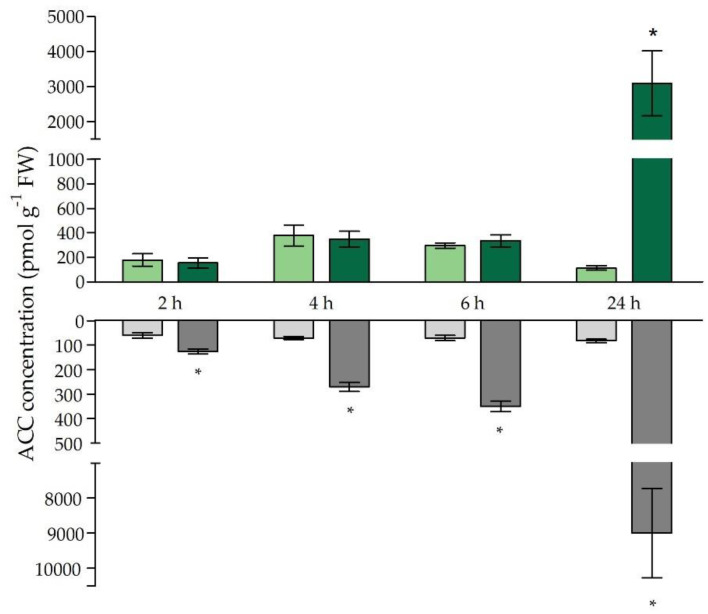
Free 1-aminocyclopropane-1-carboxylic acid (ACC) concentrations (pmol g^−1^ fresh weight) in leaves (green bars) and roots (grey bars) of *Arabidopsis thaliana* plants grown under control conditions (0 μM CdSO_4_, light bars) or exposed to 5 μM CdSO_4_ (dark bars) during 2 h, 4 h, 6 h and 24 h after 3 weeks of growth. For each time point, data represent the mean ± S.E. of four biological independent replicates. Significant differences (*t*-test: *p* < 0.05) between control and exposed plants, within each time point, are indicated with an asterisk (*).

**Figure 5 ijms-21-06232-f005:**
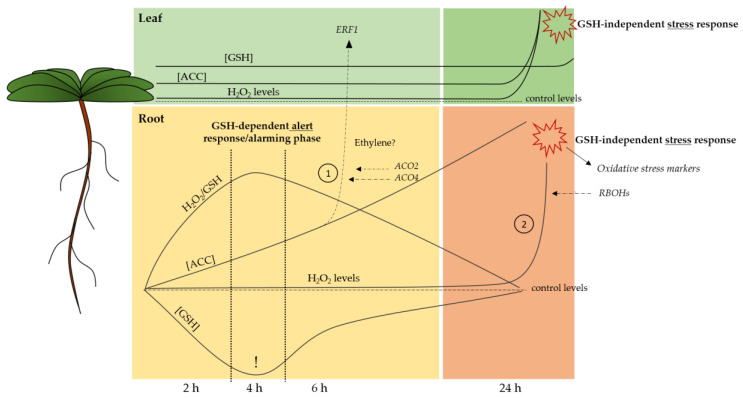
Proposed spatiotemporal timing for the identified pressure points that precede acclimation under cadmium (Cd) stress. (1) Roots serve as command centre, possibly via ethylene, to engage the leaves in an optimal response later on. (2) Stress persistence: responses are intensified via the increased expression of *RBOHs* and elevation of H_2_O_2_. This stress response is independent of glutathione (GSH). Full lines indicate Cd-induced responses and dashed lines represent control levels. Dashed arrows refer to gene expression that possibly contributes to the Cd-induced response. Full arrows refer to Cd-induced changes in gene expression as a result of the Cd-induced response. ACC: 1-aminocyclopropane-1-carboxylic acid; H_2_O_2_: hydrogen peroxide; *ERF1:* ethylene responsive transcription factor 1: *ACO*: ACC oxidase; *RBOH:* respiratory burst oxidase homologue.

**Table 1 ijms-21-06232-t001:** Cadmium concentrations (mg kg^−1^ dry weight) in leaves and roots and the translocation factor (%) of *Arabidopsis thaliana* plants grown under control conditions (0 μM CdSO_4_) or exposed to 5 μM CdSO_4_ during 2 h, 4 h, 6 h, 24 h after 3 weeks of growth. For each time point, data represent the mean ± S.E. of four biological independent replicates. Significant differences (*t*-test: *p* < 0.05) between control and exposed plants, within each time point, are marked in green. Significant differences over time are indicated with different letters (one-way ANOVA: *p* < 0.05). LOD: Cd levels below limit of detection (<10 ppb).

**Cd Concentration (mg kg^−1^ DW)**					
**Organ**	**[CdSO_4_]**	**2 h**	**4 h**	**6 h**	**24 h**
*Leaf*	0 µM	0.68 ± 0.12	0.62 ± 0.16	0.65 ± 0.09	0.90 ± 0.17
	5 µM	6.86 ± 0.17 ^a^	61.01 ± 2.73 ^b^	152.97 ± 4.84 ^c^	840.24 ± 11.71 ^d^
*Root*	0 µM	LOD	LOD	LOD	LOD
	5 µM	536.44 ± 14.24 ^a^	620.39 ± 13.40 ^b^	730.78 ± 14.25 ^c^	1788.57 ± 65.40 ^d^*
**Translocation Factor**					
	**[CdSO_4_]**	**2 h**	**4 h**	**6 h**	**24 h**
	5 µM	1.28 ± 0.04 ^a^	9.86 ± 0.53 ^b^	21.00 ± 0.97 ^c^	47.24 ± 1.88 ^d^

**Table 2 ijms-21-06232-t002:** Transcript levels of glutathione (GSH)-related genes in leaves and roots of *Arabidopsis thaliana* plants grown under control conditions (0 μM CdSO_4_) or exposed to 5 μM CdSO_4_ during 2 h, 4 h, 6 h and 24 h after 3 weeks of growth. For each time point, data are given as the mean ± S.E. of four biological replicates relative to the control set at 1.00. Significant differences (*t*-test: *p* < 0.05) between control and exposed plants, within each time point, are marked in colour (upregulated: 
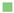
). *GSH1*: γ-glutamylcysteine synthetase; *GSH2*: GSH synthetase; *GR1*: glutathione reductase; *GGT1*: γ-glutamyl transpeptidase 1; *ZAT6*: zinc finger of *Arabidopsis thaliana* 6.

GSH-Related Genes
Gene	[CdSO_4_]	2 h	4 h	6 h	24 h
***Leaf***					
*GSH1*	0 µM	1.00 ± 0.16	1.00 ± 0.06	1.00 ± 0.13	1.00 ± 0.04
5 µM	0.75 ± 0.13	0.91 ± 0.05	0.79 ± 0.12	1.77 ± 0.19
*GSH2*	0 µM	1.00 ± 0.06	1.00 ± 0.04	1.00 ± 0.06	1.00 ± 0.04
5 µM	0.77 ± 0.11	0.91 ± 0.10	0.90 ± 0.06	2.74 ± 0.14
*GR1*	0 µM	1.00 ± 0.04	1.00 ± 0.07	1.00 ± 0.20	1.00 ± 0.10
5 µM	0.81 ± 0.08	1.05 ± 0.08	0.87 ± 0.13	3.15 ± 0.52
*GGT1*	0 µM	1.00 ± 0.26	1.00 ± 0.06	1.00 ± 0.27	1.00 ± 0.10
5 µM	0.82 ± 0.12	1.34 ± 0.19	1.47 ± 0.43	4.60 ± 0.85
*ZAT6*	0 µM	1.00 ± 0.20	1.00 ± 0.22	1.00 ± 0.08	1.00 ± 0.30
5 µM	0.93 ± 0.14	1.14 ± 0.28	1.55 ± 0.09	15.06 ± 0.55
***Root***					
*GSH1*	0 µM	1.00 ± 0.22	1.00 ± 0.16	1.00 ± 0.10	1.00 ± 0.17
5 µM	2.08 ± 0.45	1.35 ± 0.05	1.63 ± 0.06	4.67 ± 0.28
*GSH2*	0 µM	1.00 ± 0.15	1.00 ± 0.11	1.00 ± 0.01	1.00 ± 0.09
5 µM	1.67 ± 0.01	1.31 ± 0.11	1.44 ± 0.08	2.85 ± 0.20
*GR1*	0 µM	1.00 ± 0.13	1.00 ± 0.02	1.00 ± 0.04	1.00 ± 0.04
5 µM	1.01 ± 0.09	0.88 ± 0.07	1.36 ± 0.03	2.06 ± 0.19
*GGT1*	0 µM	1.00 ± 0.16	1.00 ± 0.14	1.00 ± 0.07	1.00 ± 0.13
5 µM	1.72 ± 0.11	1.67 ± 0.12	2.00 ± 0.26	5.07 ± 0.02
*ZAT6*	0 µM	1.00 ± 0.16	1.00 ± 0.13	1.00 ± 0.16	1.00 ± 0.15
5 µM	1.10 ± 0.19	1.29 ± 0.21	1.94 ± 0.02	2.89 ± 0.55

**Table 3 ijms-21-06232-t003:** Glutathione reductase (GR) activity (mU g^−1^ fresh weight) in leaves and roots of *Arabidopsis thaliana* plants grown under control conditions (0 µM CdSO_4_) or exposed to 5 μM CdSO_4_ during 2 h, 4 h, 6 h and 24 h after 3 weeks of growth. For each time point, data represent the mean ± S.E. of four biological independent replicates. Significant differences (*t*-test: *p* < 0.05) between control and exposed plants, within each time point, are marked in colour (increased: 
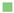
, increased: 
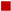
).

GR activity (mU g^−1^ FW)
[CdSO_4_]	2 h	4 h	6 h	24 h
***Leaf***
0 µM	64.46 ± 1.75	68.08 ± 0.08	65.33 ± 0.93	57.02 ± 1.55
5 µM	64.52 ± 0.86	65.70 ± 1.94	70.48 ± 1.53	69.70 ± 4.62
***Root***
0 µM	33.82 ± 1.01	32.23 ± 1.67	31.50 ± 1.26	30.56 ± 0.11
5 µM	30.16 ± 0.86	29.60 ± 0.74	30.91 ± 0.57	39.05 ± 0.93

**Table 4 ijms-21-06232-t004:** Transcript levels of prominent NADPH oxidases and oxidative stress-related genes in leaves and roots of *Arabidopsis thaliana* plants grown under control conditions (0 μM CdSO_4_) or exposed to 5 μM CdSO_4_ during 2 h, 4 h, 6 h and 24 h after 3 weeks of growth. For each time point, data are given as the mean ± S.E. of four biological replicates relative to the control set at 1.00. Significant differences (*t*-test: *p* < 0.05) between control and exposed plants, within each time point, are marked in colour (upregulated: 
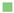
, downregulated: 
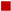
). *RBOH*: respiratory burst oxidase homologue; *ZAT12*: zinc finger of *Arabidopsis thaliana* 12; *RRTF1*: redox-responsive transcription factor 1.

NADPH Oxidases
Gene	[CdSO_4_]	2 h	4 h	6 h	24 h
***Leaf***					
*RBOHC*	0 µM	1.00 ± 0.05	1.00 ± 0.06	1.00 ± 0.14	1.00 ± 0.25
5 µM	1.01 ± 0.26	10.33 ± 4.58	3.35 ± 1.93	56.48 ± 8.23
*RBOHD*	0 µM	1.00 ± 0.25	1.00 ± 0.08	1.00 ± 0.12	1.00 ± 0.07
5 µM	0.62 ± 0.02	1.29 ± 0.22	1.60 ± 0.28	1.61 ± 0.13
*RBOHF*	0 µM	1.00 ± 0.11	1.00 ± 0.10	1.00 ± 0.18	1.00 ± 0.08
5 µM	1.40 ± 0.02	1.04 ± 0.13	1.18 ± 0.30	2.53 ± 0.55
*Root*					
*RBOHC*	0 µM	1.00 ± 0.10	1.00 ± 0.11	1.00 ± 0.14	1.00 ± 0.23
5 µM	1.41 ± 0.11	1.65 ± 0.07	1.48 ± 0.09	0.72 ± 0.08
*RBOHD*	0 µM	1.00 ± 0.06	1.00 ± 0.14	1.00 ± 0.10	1.00 ± 0.14
5 µM	1.63 ± 0.09	1.74 ± 0.13	1.60 ± 0.05	2.48 ± 0.35
*RBOHF*	0 µM	1.00 ± 0.12	1.00 ± 0.10	1.00 ± 0.05	1.00 ± 0.17
5 µM	1.74 ± 0.09	1.54 ± 0.09	1.34 ± 0.07	2.53 ± 0.14
**Oxidative Stress Markers and Related Genes**
***Leaf***					
*AT1G05340*	0 µM	1.00 ± 0.05	1.00 ± 0.04	1.00 ± 0.15	1.00 ± 0.18
5 µM	1.67 ± 0.73	1.95 ± 0.17	2.18 ± 0.68	53.23 ± 12.52
*AT1G19020*	0 µM	1.00 ± 0.30	1.00 ± 0.08	1.00 ± 0.13	1.00 ± 0.18
5 µM	1.19 ± 0.57	5.22 ± 1.97	3.20 ± 0.32	36.20 ± 7.13
*AT1G57630*	0 µM	1.00 ± 0.07	1.00 ± 0.07	1.00 ± 0.08	1.00 ± 0.13
5 µM	1.78 ± 0.97	10.67 ± 5.73	2.45 ± 1.31	30.33 ± 3.42
*AT2G21640*	0 µM	1.00 ± 0.05	1.00 ± 0.07	1.00 ± 0.09	1.00 ± 0.21
5 µM	0.83 ± 0.06	1.09 ± 0.17	0.61 ± 0.11	8.42 ± 1.33
*AT2G43510*	0 µM	1.00 ± 0.02	1.00 ± 0.08	1.00 ± 0.20	1.00 ± 0.12
5 µM	0.64 ± 0.14	1.25 ± 0.33	0.88 ± 0.14	26.36 ± 4.00
*ZAT12*	0 µM	1.00 ± 0.32	1.00 ± 0.17	1.00 ± 0.15	1.00 ± 0.40
5 µM	0.82 ± 0.33	3.26 ± 1.40	0.66 ± 0.15	18.33 ± 1.17
*RRTF1*	0 µM	1.00 ± 0.41	1.00 ± 0.48	1.00 ± 0.30	1.00 ± 0.53
5 µM	0.25 ± 0.07	0.55 ± 0.14	3.54 ± 2.25	4.59 ± 1.11
***Root***					
*AT1G05340*	0 µM	1.00 ± 0.35	1.00 ± 0.12	1.00 ± 0.35	1.00 ± 0.12
5 µM	1.11 ± 0.18	1.33 ± 0.25	0.47 ± 0.07	1.50 ± 0.31
*AT1G19020*	0 µM	1.00 ± 0.27	1.00 ± 0.14	1.00 ± 0.27	1.00 ± 0.31
5 µM	2.20 ± 0.52	1.88 ± 0.35	0.52 ± 0.09	3.56 ± 0.75
*AT1G57630*	0 µM	1.00 ± 0.10	1.00 ± 0.18	1.00 ± 0.01	1.00 ± 0.32
5 µM	1.20 ± 0.03	1.43 ± 0.26	0.46 ± 0.07	2.57 ± 0.76
*AT2G21640*	0 µM	1.00 ± 0.07	1.00 ± 0.19	1.00 ± 0.00	1.00 ± 0.11
5 µM	1.23 ± 0.05	0.95 ± 0.07	0.76 ± 0.10	1.32 ± 0.29
*AT2G43510*	0 µM	1.00 ± 0.69	1.00 ± 0.46	1.00 ± 0.32	1.00 ± 0.26
5 µM	0.60 ± 0.20	0.31 ± 0.04	0.42 ± 0.07	2.96 ± 1.38
*ZAT12*	0 µM	1.00 ± 0.04	1.00 ± 0.21	1.00 ± 0.22	1.00 ± 0.21
5 µM	1.35 ± 0.43	1.24 ± 0.31	1.09 ± 0.14	10.17 ± 2.24
*RRTF1*	0 µM	1.00 ± 0.11	1.00 ± 0.46	1.00 ± 0.69	1.00 ± 0.43
5 µM	1.27 ± 0.19	0.64 ± 0.13	1.20 ± 0.50	8.58 ± 0.66

**Table 5 ijms-21-06232-t005:** Transcript levels of ethylene-related genes in roots and leaves of *Arabidopsis thaliana* plants grown under control conditions (0 μM CdSO_4_) or exposed to 5 μM CdSO_4_ during 2 h, 4 h, 6 h and 24 h. For each time point, data are given as the mean ± S.E. of four biological replicates relative to the control set at 1.00. Significant differences (*t*-test: *p* < 0.05) between control and exposed plants, within each time point, are marked in colour (upregulated: 
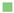
, downregulated: 
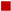
). *ACS*: ACC synthase; *ACO*: ACC oxidase; *ERF1*: ethylene responsive factor 1; *OXI1*: oxidative signal inducible 1; *MPK*: mitogen-activated protein kinase; *WRKY33*: WRKY DNA-binding protein 33.

Ethylene-Related Genes
Gene	CdSO_4_	2 h	4 h	6 h	24 h
***Leaf***					
*ACS2*	0 µM	1.00 ± 0.08	1.00 ± 0.33	1.00 ± 0.27	1.00 ± 0.18
5 µM	1.17± 0.56	2.01 ± 0.82	1.57 ± 0.21	366.12 ± 25.94
*ACS6*	0 µM	1.00 ± 0.06	1.00 ± 0.09	1.00 ± 0.02	1.00 ± 0.12
5 µM	0.91 ± 0.08	2.03 ± 0.61	3.73 ± 1.07	9.36 ± 1.76
*ACO2*	0 µM	1.00 ± 0.08	1.00 ± 0.08	1.00 ± 0.23	1.00 ± 0.10
5 µM	0.74 ± 0.09	1.08 ± 0.08	1.00 ± 0.25	5.48 ± 0.19
*ACO4*	0 µM	1.00 ± 0.06	1.00 ± 0.02	1.00 ± 0.23	1.00 ± 0.08
5 µM	0.95 ± 0.22	1.41 ± 0.05	1.76 ± 0.44	9.36 ± 2.24
*ERF1*	0 µM	1.00 ± 0.27	1.00 ± 0.24	1.00 ± 0.10	1.00 ± 0.10
5 µM	0.83 ± 0.35	2.59 ± 1.33	9.00 ± 1.09	142.90 ± 21.59
*OXI1*	0 µM	1.00 ± 0.35	1.00 ± 0.41	1.00 ± 0.27	1.00 ± 0.51
5 µM	1.07 ± 0.34	2.07 ± 0.33	10.66 ± 1.41	58.15 ± 18.54
*MPK3*	0 µM	1.00 ± 0.02	1.00 ± 0.19	1.00 ± 0.08	1.00 ± 0.06
5 µM	1.13 ± 0.13	1.53 ± 0.38	1.85 ± 0.13	4.02 ± 0.43
*MPK6*	0 µM	1.00 ± 0.04	1.00 ± 0.09	1.00 ± 0.05	1.00 ± 0.02
5 µM	1.39 ± 0.01	1.45 ± 0.09	1.00 ± 0.07	2.04 ± 0.14
*WRKY33*	0 µM	1.00 ± 0.02	1.00 ± 0.20	1.00 ± 0.07	1.00 ± 0.16
5 µM	1.30 ± 0.27	2.65 ± 0.78	3.87 ± 1.43	13.43 ± 0.93
***Root***					
*ACS2*	0 µM	1.00 ± 0.12	1.00 ± 0.07	1.00 ± 0.22	1.00 ± 0.11
5 µM	1.13 ± 0.11	0.78 ± 0.11	0.75 ± 0.08	3.76 ± 1.22
*ACS6*	0 µM	1.00 ± 0.02	1.00 ± 0.07	1.00 ± 0.14	1.00 ± 0.09
5 µM	1.61 ± 0.16	1.72 ± 0.17	1.60 ± 0.17	5.25 ± 1.83
*ACO2*	0 µM	1.00 ± 0.21	1.00 ± 0.13	1.00 ± 0.09	1.00 ± 0.31
5 µM	1.11 ± 0.17	1.15 ± 0.07	2.23 ± 0.11	5.05 ± 0.34
*ACO4*	0 µM	1.00 ± 0.17	1.00 ± 0.11	1.00 ± 0.15	1.00 ± 0.11
5 µM	1.18 ± 0.10	1.51 ± 0.14	1.95 ± 0.12	7.39 ± 1.23
*ERF1*	0 µM	1.00 ± 0.37	1.00 ± 0.33	1.00 ± 0.17	1.00 ± 0.02
5 µM	1.27 ± 0.13	1.39 ± 0.20	3.01 ± 0.50	22.01 ± 2.75
*OXI1*	0 µM	1.00 ± 0.08	1.00 ± 0.07	1.00 ± 0.12	1.00 ± 0.14
5 µM	1.27 ± 0.13	1.70 ± 0.11	1.22 ± 0.10	1.20 ± 0.37
*MPK3*	0 µM	1.00 ± 0.12	1.00 ± 0.02	1.00 ± 0.07	1.00 ± 0.12
5 µM	1.68 ± 0.18	1.51 ± 0.10	1.31 ± 0.07	2.89 ± 0.36
*MPK6*	0 µM	1.00 ± 0.14	1.00 ± 0.06	1.00 ± 0.07	1.00 ± 0.02
5 µM	1.63 ± 0.13	1.11 ± 0.02	2.93 ± 0.26	2.36 ± 0.21
*WRKY33*	0 µM	1.00 ± 0.30	1.00 ± 0.09	1.00 ± 0.23	1.00 ± 0.16
5 µM	1.15 ± 0.09	1.93 ± 0.02	0.97 ± 0.04	1.74 ± 0.44

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
