# Peer review of "Identifying the Pressure Points of Acute Cadmium Stress Prior to Acclimation in Arabidopsis thaliana"

_ijms, 2020, doi:10.3390/ijms21176232_

Round 1
Reviewer 1 Report
The authors have submitted a well researched, and well written article, it was a lovely read – thank you.
I have some minor considerations and “curiosities” for the authors, that should be considered, but not imperative to be included for publication.
Curiosities
What is the “regular level” of Cd before it becomes toxic?
How often is acute Cd stress seen in soils?
I do realise the authors are dissecting the Cd stress response through acute measures, how does this relate to plants in a wider context, i.e crops? Breeding strategy? Crispr/gm/tilling targets? Would be nice to see this addressed, particularly as heavy metal stress is a problem in some geographical areas. Not necessary for publication, but nice if it could be included somewhere.
Author Response
The authors have submitted a well researched, and well written article, it was a lovely read – thank you.
I have some minor considerations and “curiosities” for the authors, that should be considered, but not imperative to be included for publication.
Curiosities
(1) What is the “regular level” of Cd before it becomes toxic?
Cadmium (Cd) is a non-essential, toxic element. This means that, in contrast to micronutrients like Zn, Cd is considered toxic at first encounter. In our exposure set-up we used a 5 µM Cd concentration, which resembles the concentration measured in the pore water of polluted areas in Belgium (Krznaric et al. 2009). Cadmium present in the pore water is considered the bioavailable fraction and therefore, in our study, it is directly applied to the nutrient solution. Kabata-Pendias and Pendias (1992) defined 5-10 mg kg1 DW as a phytotoxicity threshold in sensitive crop species. The national legislation imposes a toxicological threshold value of 0.2 mg kg-1 FW in foliaceous vegetables like spinach and a threshold of 0.1 mg kg-1 FW in root crops like potato. Moreover, Cd is classified as a class I carcinogen by the International Agency for Research on Cancer (IARC).
(2) How often is acute Cd stress seen in soils?
In the field, Cd pollution is not homogenously distributed in the soil and because it is highly mobile, the metal is often leached out from the upper soil layer (Remans et al. 2012). For this reason, germination of seeds often takes place in absence of Cd while, upon development and growth of the roots, the seedling encounters Cd later on during its life cycle. Therefore, we applied Cd after three weeks of growth when the plant roots have grown to the extent of approximately 10 cm. In general, we consider acute stress as the stress that precedes chronical stress, which is associated with prolonged exposure and acclimation.
(3) I do realise the authors are dissecting the Cd stress response through acute measures, how does this relate to plants in a wider context, i.e crops? Breeding strategy? Crispr/gm/tilling targets? Would be nice to see this addressed, particularly as heavy metal stress is a problem in some geographical areas. Not necessary for publication, but nice if it could be included somewhere.
By means of dissecting the Cd stress response we want to unravel the molecular mechanisms plants rely on when grown on Cd-polluted soils. When the key determinants of the responses to Cd are defined, we plan to study their role and interactions using mutant plants and eventually extrapolate this knowledge to non-food crops where a more targeted approach is needed to ensure feasibility. The overall context is to ensure optimal crop production on Cd-polluted soils either by immobilising the toxic element in the soil while still being able to use these marginal lands for the cultivation of biomass or aiming at phytoextraction. We agree with the reviewer that putting our objective into a wider context is an added value for this manuscript. Therefore, we added this wider context to the first paragraph of the introduction (L30–35).
“The study of short-term plant responses allows to identify the pressure points of a certain stressor and the early challenges that plants face prior to acclimation. Understanding these early stress-induced responses will aid to improve plant acclimation itself, allowing plants, and in particular crops, to reach their full potential even in suboptimal environments. The latter can be achieved by means of biotechnological and agro-ecological approaches, which encompass o.a. genetic modifications and application of soil amendments respectively.”
Reviewer 2 Report
In this work, authors carried out a time-course (2, 4, 6 and 24 h) analysis of Cd-induced responses in A. thaliana plants grown in hydroponic culture and treated with 5 mM CdSO4 vs controls. They then evaluated: growth responses (only in terms of FW, which is very limited), root/leaf Cd accumulation (data obtained in hydroponic culture has little relevance to crops grown in soil), GSH levels, expression of GSH genes (GSH1, GSH2 and GR1), transcriptional profile of ROS-producing NADPH oxidases (RBOHD and RBOHF), oxidative stress markers and related genes (ZAT12 and RRTF1), H2O2 levels, transcriptional profiles of ethylene biosynthesis and responsive genes, and levels of the ethylene precursor ACC.
In the Abstract, the significance of this work is described as “Knowledge on Cd-induced stress responses is required to optimise cultivation of non-food crops on Cd-polluted soils”. Why then use A. thaliana, which is not a crop. Of course, it is an excellent model plant for studies on genetic control of G&D/stress responses but not ideal for this study, especially since the full potential of working with Arabidopsis was not exploited (e.g., mutants). It would have been more interesting if the study had been carried out using a non-model plant with potential use as a (non-food) crop.
Moreover, in my opinion, the work is too descriptive and does not provide any novel information on plant responses to metal stress, insofar as the involvement of GSH, ROS, and ethylene is already well documented. As stated by the authors themselves “several studies have indicated that a rapid and transient depletion of GSH occurs after exposure to excess metal concentrations”. The interesting part is that the initial responses and the later (24 h) acclimation responses seem to differ, but this is not a very novel finding either.
I also believe that there is too much speculation about the significance of the data.
Examples:
- “the allocation of GSH to PC synthesis was reflected by a significant depletion of GSH in the roots after 2 h of exposure”. There is no evidence for this (not even correlative data) since PCs were not determined.
- “Therefore, it can be hypothesised that GGT encoded by GGT1 also functions as an important modulator in the redox sensing and signalling under Cd stress.”
- “Moreover, we hypothesise that, corresponding to other studies, the root GSH depletion is a requirement for a proper Cd-induced alert response and, consequently, optimal acclimation.”
- “Finally, we hypothesise that the GSH-based alarming phase will aid to enhance root ethylene synthesis, derived from the rapidly increased ACC concentrations. This root-derived ethylene might function as a root-to-shoot signal that serves to optimise responses in the leaves, which eventually result in increased leaf GSH levels.”
Based on all these considerations, I think the manuscript should be rejected but could be submitted to another journal as the work was correctly performed and the data are well presented.
Author Response
In general, we appreciate the concerns of reviewer 2 and similar as mentioned to the editor, we rephrased the overall aim of this study. As such the focus of this study is better addressed (and also clearly linked to the topic of this special issue) and less confusion with the wider context is avoided.
Remark 1 & 2:
In this work, authors carried out a time-course (2, 4, 6 and 24 h) analysis of Cd-induced responses in A. thaliana plants grown in hydroponic culture and treated with 5 mM CdSO4 vs controls. They then evaluated: growth responses (remark 1: only in terms of FW, which is very limited), root/leaf Cd accumulation (remark 2: data obtained in hydroponic culture has little relevance to crops grown in soil), GSH levels, expression of GSH genes (GSH1, GSH2 and GR1), transcriptional profile of ROS-producing NADPH oxidases (RBOHD and RBOHF), oxidative stress markers and related genes (ZAT12 and RRTF1), H2O2 levels, transcriptional profiles of ethylene biosynthesis and responsive genes, and levels of the ethylene precursor ACC.
Revision remark 1 & 2
- Extensive research on the growth responses of thaliana under 5 µM of Cd exposure was already performed within our research group. It was shown by Keunen et al. (2011) that exposure to 5 µM of Cd had no effect on the growth of A. thaliana, more specifically on leaf development and rosette radius, after 24 h of exposure. Schellingen et al. (2014) indicated that no root and leaf growth inhibition was observed after 24 h. Similarly, Cuypers et al. (2011) demonstrated the lack of toxicity symptoms and growth reduction after 24 h exposure to 5 µM of Cd. This was also observed in our study. Therefore, we did not consider it an added value to further focus on additional growth parameters. Similar to the fresh weight, we did not observe significant effects on the determined dry weight, we included these additional data in the supplementary materials.
“Fresh weight (Figure 1), dry weight (Supplemental Figure S1) and Cd concentration (Table 1) were compared between Cd-exposed and unexposed WT plants within the short exposure time frame. Acute exposure did not have a negative impact on the root and leaf fresh weight and dry weight of WT A. thaliana plants (Figure 1 and Supplemental Figure S1).”
- We understand that the use of a hydroponics cultivation system cannot be directly translated to the field. However, the major aim of this study was to further obtain fundamental knowledge on Cd-induced responses that aid a more targeted, and therefore feasible, approach in future crop studies. A hydroponics cultivation method was used to enable a controlled exposure to Cd while other parameters were kept constant. In this way, we could specifically focus on the Cd-induced effects in both roots and leaves. Moreover, the use of a hydroponics system enabled us to harvest roots in an efficient and precise way, which is often difficult when roots are harvested from soil-grown plants. Furthermore, by applying 5 µM Cd to the nutrient solution, we mimicked the concentration of 5 µM Cd measured in the pore water of polluted areas, which is considered the bioavailable Cd fraction. The use of hydroponics enabled a precise administration of 5 µM to the roots system, circumventing the buffering capacity of soil. Furthermore, in the field Cd is not homogenously distributed and most of the times seed germination occurs in absence of Cd. Consequently, the pollutant is encountered later on when seedlings have grown a sufficient amount and the root system reaches the lower layers of the soil. For this reason, we administered Cd to the nutrient solution not at the beginning of the growth period but when the roots had grown a sufficient length.
Remark 3
In the Abstract, the significance of this work is described as “Knowledge on Cd-induced stress responses is required to optimise cultivation of non-food crops on Cd-polluted soils”. Why then use A. thaliana, which is not a crop. Of course, it is an excellent model plant for studies on genetic control of G&D/stress responses but not ideal for this study, especially since the full potential of working with Arabidopsis was not exploited (e.g., mutants). It would have been more interesting if the study had been carried out using a non-model plant with potential use as a (non-food) crop.
Revision remark 3
We realise and agree with the reviewer that this sentence may confuse the reader in expecting that the manuscript covers translational research. However, we are aware that our research is fundamental and that this sentence was misleading, as we never intended to depict our data as translational.
Throughout the manuscript we better addressed the focus of our study and how this can be seen and studied in a wider framework (also suggested by reviewer 1 - suggestion 3). Therefore, we made the following adaptations in the abstract and introduction:
Abstract: “The toxic metal cadmium (Cd) is a major soil pollutant. Knowledge on the acute Cd-induced stress responses is required to better understand the triggers and sequence of events that precede plant acclimation.”
Introduction: “. The study of short-term plant responses allows to identify the pressure points of a certain stressor and the early challenges that plants face prior to acclimation. Understanding the early stress-induced responses will aid to improve plant acclimation itself, allowing plants, and in particular crops, to reach their full potential even in suboptimal environments. The latter can be achieved by means of biotechnological and agro-ecological approaches which encompass a.o. genetic modifications and application of soil amendments respectively.”
We find that this fundamental research in A. thaliana is valuable to clearly outline the spatiotemporal responses of key regulators in A. thaliana to Cd exposure. This is required on one hand to allow a more targeted study using mutant plants to further fundamentally unravel the underlying signalling mechanisms. On the other hand, targeted analyses can be used when considering crops, which are more labour-intensive to study. Therefore, we think it is necessary to share these findings with other researchers.
Remark 4
Moreover, in my opinion, the work is too descriptive and does not provide any novel information on plant responses to metal stress, insofar as the involvement of GSH, ROS, and ethylene is already well documented. As stated by the authors themselves “several studies have indicated that a rapid and transient depletion of GSH occurs after exposure to excess metal concentrations”. The interesting part is that the initial responses and the later (24 h) acclimation responses seem to differ, but this is not a very novel finding either.
Revision remark 4
Indeed, it was already demonstrated that GSH depletion occurs after 2 h of Cd by our research group due to its allocation to PC synthesis. However, the behaviour of the metabolite over time was not elucidated yet. For example, the even stronger GSH depletion observed after 4 h and the ongoing recovery after 6 h are new insights. As mentioned by reviewer 2, ROS, ethylene and GSH are known to be involved in Cd stress. However, the goal of this study was to further elucidate how these regulators are involved, what the timing is of their involvement and what their importance is. We are convinced that this manuscript covers a range of novel insights and that these data are significant to our research field. Some overlaps with previous studies are indeed mentioned but this, in our opinion, further substantiates the data as its reproducibility is proven and this concerns only a small subset of the described data. Compiling this new dataset, and confirmation with previous data (other time points), enabled us to compile a spatiotemporal model of cadmium signalling.
Novelties presented in the manuscript:
- Time course analysis of GSH-related responses in both root and leaf (with the response after 4 h and 6 h as new insights)
- Time course Cd determination in both root and leaf
- Time course gene expression analysis in both root and leaf
- Time course determination of GR activity in both root and leaf
- Time course determination of H2O2 in both root and leaf
- The GSH/H2O2 ratio as a measure for the plant’s redox state over time
- Time course determination of ACC content in both root and leaf
- Strong transcriptional induction of GGT1 implementing its role in the Cd-induced stress response
- Spatiotemporal model that shows the sequence of events concerning key regulators in the Cd-induced acute stress response
Remark 5
I also believe that there is too much speculation about the significance of the data.
Examples:
Remark 5a
“the allocation of GSH to PC synthesis was reflected by a significant depletion of GSH in the roots after 2 h of exposure”. There is no evidence for this (not even correlative data) since PCs were not determined.
Revision remark 5a
This was previously determined by Jozefczak et al. (2014) using plants grown under the same cultivation conditions.
“In this study, the allocation of GSH to PC synthesis was reflected by a significant depletion of GSH in the roots after 2 h of exposure (Figure 2), which was also demonstrated by Jozefczak et al. (2014).”
We agree with the reviewer that our discussion should be written in a more direct way to better address the main message of our study that is fully supported by the present and previously obtained data.
Remark 5b
L338-339
“Therefore, it can be suggested that GGT encoded by GGT1 also functions as an important modulator in the redox sensing and signalling under Cd stress.”
Remark 5c
L416-417
“Moreover, we conclude that, corresponding to other studies, the root GSH depletion is a requirement for a proper Cd-induced alert response and, consequently, optimal acclimation.”
Remark 5d
L428-430
“Finally, we put forward a model wherein the GSH-based alarming phase will aid to enhance root ACC levels and ethylene synthesis, derived from the rapidly increased ACC concentration (Figure 5). This root-derived ethylene and/or ACC might function as a root-to-shoot signal that serves to optimise responses in the leaves, which eventually result in increased leaf GSH levels.”
In addition, we applied this writing style also at multiple other places in the discussion (see manuscript with track changes).
Remark 6
Based on all these considerations, I think the manuscript should be rejected but could be submitted to another journal as the work was correctly performed and the data are well presented.
We thank the reviewers for their valuable comments and by better addressing the main goal and hence also the novelty of our study, we believe that we improved this manuscript for publication in the special issue “ROS and Abiotic Stress in Plants” of IJMS.
Round 2
Reviewer 2 Report
Following my recommendations, the authors have improved the manuscript both in terms of the rationale of the study (by eliminating any emphasis on crops) and discussion of the results. In its present form it is acceptable for publication.